# Clinical Characteristics and Outcomes of COVID-19 Hospitalized Patients: A Comparison between Complete mRNA Vaccination Profile and Natural Immunity

**DOI:** 10.3390/jpm12020259

**Published:** 2022-02-10

**Authors:** Iosif Marincu, Cosmin Citu, Felix Bratosin, Iulia Bogdan, Madalina Timircan, Camelia Vidita Gurban, Adrian Vasile Bota, Laurentiu Braescu, Mirela Loredana Grigoras

**Affiliations:** Methodological and Infectious Diseases Research Center, Department of Infectious Diseases, “Victor Babes” University of Medicine and Pharmacy, 300041 Timisoara, Romania; imarincu@umft.ro (I.M.); felix.bratosin7@gmail.com (F.B.); iulia.georgianabogdan@gmail.com (I.B.); timircan.madalina@yahoo.com (M.T.); gurban.camelia@umft.ro (C.V.G.); bota.adrian1@yahoo.com (A.V.B.); laurbraescu@gmail.com (L.B.); mirela.grigoras@yahoo.com (M.L.G.)

**Keywords:** SARS-CoV-2, COVID-19, mRNA vaccine, natural immunity, reinfection, breakthrough infection

## Abstract

Although laboratory data show that antibody responses to COVID-19 immunization give superior neutralization of certain circulating variations to spontaneous infection, few real-world epidemiological studies demonstrate the advantage of vaccination for previously infected individuals. This paper summarizes the outcomes of a case–control study conducted in Romania between March 2020 and October 2021 on patients previously infected with SARS-CoV-2. A case–control study was implemented after identification of 62 breakthrough cases. These cases were matched by age and gender to a 1:1 ratio with a control group of unvaccinated patients with SARS-CoV-2 reinfection status. There were no significant differences in the severity of cases and mortality between the study groups. However, unvaccinated patients had a shorter protection from natural immunity than patients with full vaccination status (58 days versus 89 days). The unvaccinated cases with SARS-CoV-2 reinfection were also statistically more likely to have a longer hospital admission duration (12.4 days versus 9.8 days), and required more non-invasive oxygen supplementation during their stay than breakthrough cases (37.1% versus 19.4%). Individuals with prior SARS-CoV-2 infection who were not vaccinated are not at a higher risk of severe COVID-19 infection or mortality compared to those who were completely vaccinated with the mRNA vaccine Comirnaty^®^ Pfizer/BioNTech BNT162b2 and acquired a breakthrough infection within 2–3 months of the previous infection with a Beta or Delta SARS-CoV-2 variant. Although our findings are consistent with natural immunity offering similar short-term protection to a second dose of mRNA vaccine, all eligible individuals should be provided with immunization to lower their risk of infection, even if they have already been infected with SARS-CoV-2.

## 1. Introduction

Due to a lack of efficient treatments, numerous attempts have been undertaken to avoid infection and sickness since the discovery of severe acute respiratory syndrome coronavirus 2 (SARS-CoV-2) as the causative agent of coronavirus disease 19 (COVID-19) in late 2019. After the World Health Organization declared a global pandemic in early 2020 [1], a race to develop a functional vaccine was carried out in the majority of countries possessing advanced research capabilities. In less than one year, extremely effective vaccinations were released, and a global-scale vaccination campaign began [2]. Since its launch in late 2020, the messenger RNA (mRNA) vaccine (Comirnaty^®^ Pfizer/BioNTech BNT162b2) has proved to be very successful in preventing clinically significant COVID-19 [3]. Additionally, the vaccination has been found to decrease the prevalence of asymptomatic illness and its related infectiousness [4]. In clinical studies, several approved vaccines showed great effectiveness in protecting against SARS-CoV-2 infection, ranging from 70% to 95% for the Beta variant. However, post-authorization, real-world studies are critical to determining the vaccine’s efficacy in varied groups and under uncontrolled real-world conditions [5,6]. Even so, after promising trial results, vaccination recipients have had breakthrough infections, a situation that has been documented in various nations and health care organizations [7]. There has been no published investigation of the correlation between protection against breakthrough infection and the type of vaccine.

In real-world settings, the Comirnaty^®^ Pfizer/BioNTech BNT162b2 vaccine prevents roughly 90% of SARS-CoV-2 infections and 94–100% of severe or fatal illness [8], and was proven to provide better neutralization than natural infection does [9]. Nonetheless, COVID-19 reinfection is increasing, in both vaccinated and unvaccinated individuals with natural immunity from previous infection. Under present circumstances, studies suggest that reinfection by SARS-CoV-2 is expected to occur between 3 and 51 months after peak antibody response, with a median of 16 months [10]. Reinfection seems rare after spontaneous infection, with a documented frequency of around 0.7%, as reported in the UK [11], although no data are available from Romania. Recently, breakthrough infections have been documented in fully vaccinated individuals, while little is known about the risk factors, clinical presentation, and outcomes of breakthrough infections when compared to demographically and clinically comparable controls [12].

The need for identifying and protecting those at greater risk of post-vaccination illness is growing as more people get vaccinated. Prior to the availability of vaccines, groups at increased risk of SARS-CoV-2 infection included frontline health care workers and residents of areas of greater relative deprivation, likely due to increased exposure [13], while risk factors, such as increasing age, male sex, multimorbidity, and frailty, are associated with poorer COVID-19 outcomes [14]. The timescale for reinfection is critical for a variety of public health decision-making processes. Reinfection is predicted to become more prevalent as the COVID-19 pandemic proceeds. Maintaining public health measures to limit transmission, including among previously infected patients with SARS-CoV-2, along with ongoing attempts to expedite immunization globally, are crucial for preventing COVID-19 morbidity and death.

COVID-19 patients exhibit a range of symptoms and clinical requirements [15]. Identifying risk groups for intervention, forecasting medical resource needs, and establishing appropriate testing procedures requires elucidating symptom patterns in persons with COVID-19 after immunization. Additionally, some unprotected patients who develop COVID-19 have a longer illness (so-called long COVID), and it is uncertain if immunization lessens the chance of developing long COVID [16].

After the COVID-19 vaccination campaign was deployed in Romania at the end of December 2020 [17], until October 2021, around 5 million people, or a total of 30% of the eligible population, were fully vaccinated. [18]. Yet, cases of breakthrough infection were reported more often as time passed and the fourth wave was developing. Therefore, research was conducted on a fully vaccinated population, and a naturally immunized population, to investigate lifestyle, sociodemographic, and personal risk factors for post-vaccination infection, or reinfection, as well as disease outcomes in these two populations. We sought to characterize individual risk factors for SARS-CoV-2 infection after full vaccination or reinfection in unvaccinated patients, and to compare the duration, severity, and symptom profile of illness in individuals with SARS-CoV-2 infection following their second vaccination to those who were not vaccinated, but were documented as having COVID-19 disease.

## 2. Materials and Methods

### 2.1. Design and Eligibility Criteria

We conducted a case-control study to compare a group of COVID-19 patient-cases with breakthrough infection with a group of reinfected patient-controls using demographic, clinical, and laboratory information from the database of patients hospitalized at the Infectious Diseases and Pulmonology Hospital “Dr. Victor Babes”, Timisoara, from 1 March 2020 to 1 October 2021. A total of 2041 patients with SARS-CoV-2 infection were identified in the administrative database at the Infectious Diseases and Pulmonology Hospital “Dr. Victor Babes”, Timisoara, during the study period. Among these, we identified 62 breakthrough cases, defined by SARS-CoV-2 infection in a fully vaccinated patient. In the remaining 1979 admissions without a history of COVID-19 vaccination, there were 168 cases of reinfection, of which 62 were matched by age and gender with the breakthrough cases to a 1:1 ratio.

The condition for cases and controls was the presence of antibodies for SARS-CoV-2 by the means of vaccination or prior infection. Patients aged 18 years, with proven SARS-CoV-2 infection as determined by positive reverse transcription polymerase chain reaction (RT-PCR) test findings recorded in our database within the time period specified, were eligible for inclusion.

Variables included in the study comprised: age; gender; comorbidity count (including the at-risk comorbidities as previously described [19]; BMI (Body Mass Index); smoking status; days until second infection (defined by the time elapsed since first SARS-CoV-2 infection); oxygen saturation on admission; severe infection status (defined by respiratory failure requiring mechanical ventilation, with a respiratory rate greater than 30/min, oxygen saturation of hemoglobin measured by pulse-oximetry (SpO_2_) < 90%, coagulation disorders, failure of other organs requiring admission to the intensive care unit (ICU), and ground-glass opacities involving more than 60% of the lungs on the chest X-ray or computed tomography (CT) scan); at-risk comorbidity count (including the comorbid conditions identified as independent risk factors for mortality in COVID-19: cancer, chronic kidney disease, chronic obstructive pulmonary disease, heart disease, obesity, type 2 diabetes mellitus, and immunosuppression); infection transmission (defined by the method of viral transmission, in contact with a family member living in the same household, a coworker, or unknown contact); ground glass opacities (quantified on X-ray or CT scan); oxygen supplementation requirements (AIRVO—high-flow nasal cannula, CPAP—continuous positive airway pressure, and mechanical ventilator); intensive care unit admission; mortality; and days of hospital stay.

### 2.2. Ethics

The Local Commission of Ethics for Scientific Research from the “Dr. Victor Babes” Clinical Hospital for Infectious Diseases and Pulmonology in Timisoara operates under art. provisions 167 of Law no. 95/2006, art. 28, chapter VIII of order 904/2006, with EU GCP Directives 2005/28/EC, International Conference of Harmonisation of Technical Requirements for Registration of Pharmaceuticals for Human Use (ICH), and with the Declaration of Helsinki—Recommendations Guiding Medical Doctors in Biomedical Research Involving Human Subjects. The current study was approved on the 15 December 2021, and issued with the approval number 12568.

### 2.3. Statistics

We compared COVID-19 patients with and without full vaccination status using IBM SPSS v.26 statistical software (Chicago, IL, USA). Case matching was performed by age and gender using the “Case–Control Matching” function in SPSS. The χ^2^ test and Fisher’s exact test were used to analyze proportions and Student’s t-test or Mann–Whitney U-test for continuous and categorical data, respectively. Risk analysis was performed by using a logistic regression model and Kaplan–Meyer curve. Risk factors are reported as odds ratios (ORs) with 95% confidence intervals (CIs). The significance threshold was set for α = 0.05.

## 3. Results

Between February 2020 and October 2021, approximately 1.7 million Romanians were infected with SARS-CoV-2, while more than 4 million people received a second vaccine dose since the start of the vaccination campaign in December 2020, until October 2021, with the Comirnaty^®^ Pfizer/BioNTech BNT162b2 vaccine [20]. We identified a total of 2041 patients with COVID-19 in our database, of which 62 were breakthrough cases. These cases were matched 1:1 by age and gender with cases of reinfection in unvaccinated patients.

Regarding patients’ general characteristics (Table 1), there were few significant differences identified between cases and controls. The time until second infection in reinfection cases was statistically shorter than in breakthrough cases (89.4 ± 42.3 days versus 58.6 ± 31.3 days, *p*-value < 0.001) (Figure 1). We observed a significant difference between study groups after counting the comorbidities (*p*-value = 0.044), where 42% of the breakthrough cases had more than three comorbidities identified at admission, compared to 25% in the control group. The control group, however, had 48% of individuals with 1–3 comorbidities, versus 27% in the breakthrough cases. Infection transmission in the breakthrough cases was unknown in 61% of admissions, compared to 32% in reinfection cases, where the major route of transmission was within the household (42%), (the overall proportion’s *p*-value = 0.003). There were no significant differences in lung involvement (*p*-value = 0.162), and oxygen supplementation requirements with the use of AIRVO (*p*-value = 0.288), or ventilators (*p*-value = 0.263). On the contrary, significantly more patients in the control group required ventilator support (*p*-value = 0.028). The ICU admissions did not differ significantly between groups, nor did the number of days of stay in the ICU, or the mortality. However, we observed that vaccinated patients with breakthrough infections had a significantly lower duration of hospital admission (9.8 days versus 12.4 days, *p*-value < 0.001) (Figure 2).

We identified the main complaints after hospital discharge in both study groups (Table 2). Although the unvaccinated patients reported more symptoms of anosmia and ageusia than the vaccinated patients (28% versus 14%, and 28% versus 19%), the differences were not statistically significant (*p*-value = 0.058, and 0.247, respectively).

Severe disease was identified in 17% of the breakthrough cases, compared to 25% in the unvaccinated group with natural immunity, although the difference was not significant (*p*-value = 0.276). Mortality was also not significant between cases and controls (*p*-value = 0.752). A risk assessment was performed (Table 3), identifying unvaccinated patients with natural immunity as having 1.55 greater odds of severe disease or mortality than vaccinated patients, although the risk was not proven statistically significant (*p*-value = 0.061). The age of patients was a significant risk factor (OR = 3.31, *p*-value < 0.001). Other risk factors were the number of comorbidities (*p*-value < 0.001), smoking status (*p*-value = 0.026), lung involvement > 60% (*p*-value < 0.001), and oxygen saturation < 92% (*p*-value = 0.004). The mixed model comprising all significant individual risk factors showed an odds ratio of 1.36 (CI = 1.02–3.83, *p*-value = 0.001).

## 4. Discussion

This research discovered that inhabitants of western Romania who had previously been infected with SARS-CoV-2 in 2020, and were not immunized against COVID-19, had a similar risk of reinfection to those who received a second dose of Comirnaty^®^ Pfizer/BioNTech BNT162b2 vaccine. Our findings are consistent with data from other countries regarding breakthrough cases and reinfection cases, reported to be around 1% of the total number of SARS-CoV-2 infections from the time the pandemic started [21]. Unvaccinated patients with reinfection showed a considerably increased duration of hospital stay, and increased requirements for oxygen supplementation throughout their admission. However, this result might be biased, since patients in the reinfection group had significantly more comorbidities than breakthrough cases. The risk analysis showed that unvaccinated patients were 2.55 times more likely to develop a more-severe form of COVID-19 than vaccinated patients. This study bolsters the Centers for Disease Control and Prevention (CDC)’s recommendation that all eligible individuals be administered the COVID-19 vaccine, regardless of their history of SARS-CoV-2 illness [22].

Although reinfection with SARS-CoV-2 has been established, scientific knowledge of infection-induced immunity is still developing [23]. The length of immunity resulting from spontaneous infection is still unknown, but is believed to last for 90 days in the majority of people. This hypothesis is supported by our study, which identified an average duration until reinfection in unvaccinated patients of 58 days.

The introduction of novel variants may influence the durability of infection-induced immunity. Laboratory investigations have shown that sera from previously infected individuals may have poor or inconsistent responses to a number of concerning variants [24]. For instance, a recent laboratory investigation discovered that sera taken from previously infected individuals prior to vaccination had a much lower, and in some instances non-existent, neutralization response to the B.1.351 (Beta) variant when compared to the original Wuhan-Hu-1 strain [25]. Sera from the same individuals after vaccination had an enhanced neutralizing response to the Beta variant, suggesting that vaccination improves the immune response to variants to which the infected individual had not previously been exposed. Although such laboratory research continues to imply that vaccination improves neutralization of SARS-CoV-2 variants, little evidence from real-world situations supports the conclusion that immunization may increase protection for previously infected individuals. However, the results of this research cannot indicate whether complete immunization is associated with a lower risk of reinfection in previously infected individuals, or if being unvaccinated is associated with a greater risk of reinfection.

A study of fully vaccinated US veterans found that advanced age and anemia were associated with an increased risk of post-vaccination illness, but African American persons had a lower risk than White persons [26]. In comparison, our study did follow the patients for symptomatic complaints after disease clearance, but did not assess what risk factors are responsible for post-vaccination illness, and only the Caucasian race was present in our study.

Regarding study limitations, it should be mentioned that the sample size was relatively small, including 62 breakthrough cases (defined by SARS-CoV-2 infection in a fully vaccinated patient), and 62 reinfection control-cases. These proportions should be reported at the total percentage of the vaccinated Romanian population, and the total number of COVID-19 cases at the time of study. Moreover, the number of cases we described are counted from those patients who required hospital admission, and not the overall count, making the estimated proportion of breakthrough cases unreliable. There are studies that managed to include larger sample sizes of breakthrough cases [27,28], although they lack a comparison with cases of reinfection in the presence of natural immunity after an initial infection. Another major limitation of this study is the lack of serological data to indicate whether reinfection occurred with the same SARS-CoV-2 viral strain. Lastly, it is important to mention that, at the time of study, the Beta and Delta variants were the main circulating SARS-CoV-2 variants, therefore making the results debatable with respect to the Omicron variant or other strains that were not studied yet.

## 5. Conclusions

Individuals with previous SARS-CoV-2 infection who were unvaccinated are not at increased risk of severe COVID-19 or death compared with those who were fully vaccinated with Comirnaty^®^ Pfizer/BioNTech BNT162b2 and developed a breakthrough infection, in cases of reinfection occurring within 2–3 months of the initial infection. These findings are consistent with natural immunity offering similar short-term protection to a second dose of mRNA vaccine in cases of infection with the Beta or Delta SARS-CoV-2 variants.

## Figures and Tables

**Figure 1 jpm-12-00259-f001:**
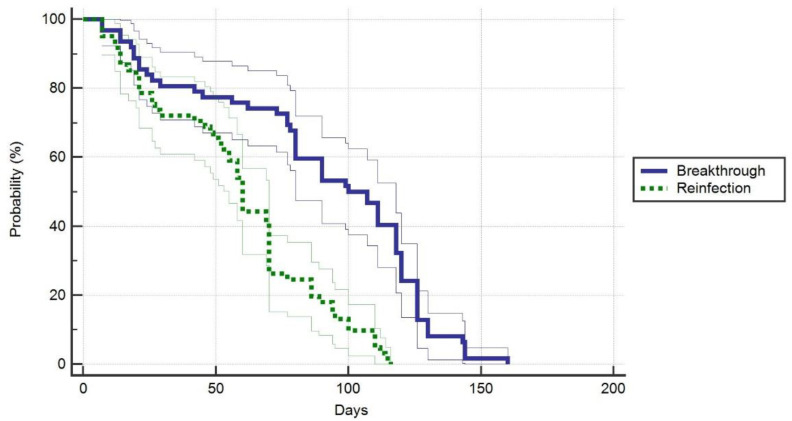
Probability of duration until breakthrough after second vaccine dose or reinfection in unvaccinated patients.

**Figure 2 jpm-12-00259-f002:**
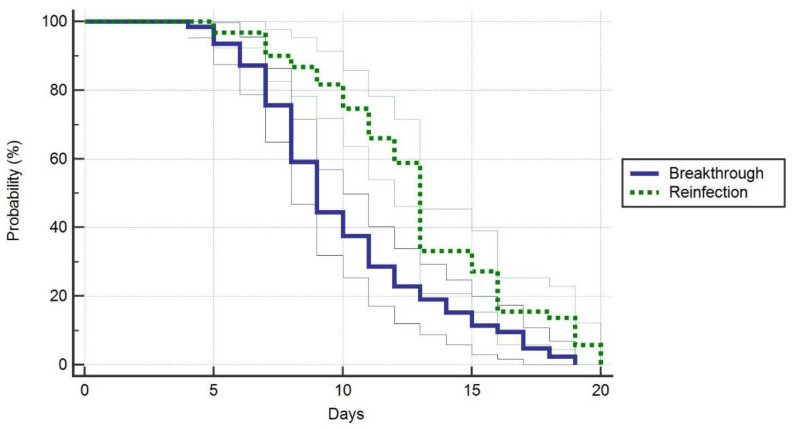
Probability of hospital stay duration in symptomatic patients.

**Table 1 jpm-12-00259-t001:** Main characteristics of the cases and controls. Case-matched by age and gender.

Characteristics	Breakthrough Cases *n* = 62	Reinfection Cases *n* = 62	*p*-Value
**General**			
Median age (IQR)—years	51 (37.5–61.5)	51 (37.5–61.5)	1
Male gender—*n* (%)	39 (62.9%)	39 (62.9%)	1
Obesity—(BMI ≥ 30 kg/m^2^)	18 (29.0%)	15 (24.1%)	0.542
Smoking, yes—*n* (%)	20 (32.2%)	17 (27.4%)	0.555
Days until second infection (mean ± SD)	89.4 ± 42.3	58.6 ± 31.3	<0.001
Oxygen saturation on admission ≤ 92%—*n* (%)	29 (46.7%)	38 (61.2%)	0.104
Severe infection—*n* (%)	11 (17.7%)	16 (25.8%)	0.276
**At-risk comorbidity count** *			0.044
None	19 (30.6%)	16 (25.8%)	
1–3 comorbidities	17 (27.4%)	30 (48.4%)	
>3 comorbidities	26 (42.0%)	16 (25.8%)	
**Infection transmission**			0.003
Family	12 (19.4%)	26 (42.0%)	
Colleagues	12 (19.4%)	16 (25.8%)	
No contact history	38 (61.2%)	20 (32.2%)	
**Ground glass opacities**			0.162
<30%	17 (27.4%)	11 (17.7%)	
30–60%	35 (56.5%)	33 (53.2%)	
>60%	10 (16.1%)	18 (29.0%)	
**Oxygen supplementation**			
AIRVO	17 (27.4%)	12 (19.4%)	0.288
CPAP	12 (19.4%)	23 (37.1%)	0.028
Ventilator	10 (16.1%)	15 (16.1%)	0.263
ICU admission	11 (17.7%)	16 (25.8%)	0.276
Days in the ICU (mean ± SD)	12.2 ± 6.7	13.4 ± 8.6	0.387
Mortality	5 (8.1%)	6 (9.6%)	0.752
Days until discharge (mean ± SD)	9.8 ± 3.4	12.4 ± 3.8	<0.001

* As described in the Materials and Methods section; IQR—Interquartile Range; BMI—Body Mass Index; SD—Standard Deviation; AIRVO—high-flow nasal cannula; CPAP—Continuous Positive Airway Pressure; ICU—Intensive Care Unit.

**Table 2 jpm-12-00259-t002:** Patient outcomes after hospital discharge.

Outcomes (*n*, %)	Breakthrough Cases *n* = 57	Reinfection Cases *n* = 56	*p*-Value
Cough	28 (49.1%)	35 (62.5%)	0.152
Rhinitis	16 (28.1%)	20 (35.7%)	0.383
Chest pain	5 (8.7%)	6 (10.7%)	0.727
Dyspnea	20 (35.1%)	17 (30.3%)	0.592
Palpitations	6 (10.5%)	5 (8.9%)	0.774
Headache	10 (17.5%)	13 (23.2%)	0.454
Fever	3 (5.2%)	3 (5.3%)	0.982
Anosmia	8 (14.0%)	16 (28.5%)	0.058
Ageusia	11 (19.2%)	16 (28.5%)	0.247
Anorexia	12 (29.0%)	10 (17.8%)	0.667
Diarrhea	3 (5.2%)	5 (8.9%)	0.447
Myalgia	14 (24.5%)	11 (19.6%)	0.528
Insomnia	21 (36.8%)	18 (32.1%)	0.599
Anxiety	18 (31.5%)	19 (33.9%)	0.790
Depression	10 (17.5%)	14 (25.0%)	0.332

**Table 3 jpm-12-00259-t003:** Risk factors associated with severe illness and mortality among breakthrough cases and reinfection cases in unvaccinated patients.

Factors	Odds Ratio (95% CI)	*p*-Value
Unvaccinated	1.55 (CI = 0.91–2.67)	0.061
Age (>median of the study)	3.31 (CI = 1.82–4.59)	<0.001
Male gender	1.04 (CI = 0.60–1.28)	0.694
Comorbidities (>3)	1.57 (CI = 1.46–3.81)	<0.001
Obesity	1.13 (CI = 0.85–1.47)	0.724
Smoking	2.33 (CI = 1.16–4.52)	0.026
Ground glass opacities (>60%)	4.09 (CI = 2.10–7.55)	<0.001
Oxygen saturation (<92%)	2.66 (CI = 1.42–3.90)	0.004
Days until second infection	1.87 (CI = 0.92–3.77)	0.088
Days in the ICU	1.60 (CI = 0.70–2.19)	0.195
Mixed model *	1.36 (CI = 1.02–3.83)	0.001

* All risk factors combined; CI—Confidence Interval; ICU—Intensive Care Unit.

## Data Availability

Data available on request from the corresponding author.

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
