# Peer review of "Clinical Characteristics and Outcomes of COVID-19 Hospitalized Patients: A Comparison between Complete mRNA Vaccination Profile and Natural Immunity"

_jpm, 2022, doi:10.3390/jpm12020259_

Round 1

Reviewer 1 Report

The paper by Marincu et. deals with the occurrence of infection in patients after vaccination and in patients who have not been vaccinated but have undergone the previous infection. It addresses important research and societal issue regarding resistance to SARS-COV-2 infection and their breakthrough. The paper is written in a clear and lucid manner, however, I have several comments to raise.

The authors report 2041 patients, 62 of whom had infection after vaccination and were matched with 62 patients who developed reinfection. Please complete the following information about the group from the database:

- how many people in the 2041 group were vaccinated, what percentage of the vaccinated group are these 62 people who breakthrough infection?

- how many people in the 2041 group were in total who passed infection without vaccination, and how many of these people had reinfection?

- Can the statement "vaccines provide better neutralization than natural infection does" (line 55-56 or in line 224-226) quoted in the paper could be illustrated from the above data?

- Was the occurrence of reinfection related to the severity of the primary infection? Did patients with severe COVID-19 differ in the occurrence of reinfection from mild or asymptomatic patients?

I have comments about the selection of the control group. Are there any publications documenting the correctness of such selection of control group? How was the requirement for randomization of the control group fulfilled, was the control group matched by some described algorithm e.g. Propensity score matching? The control group differs in comorbidities from the study group (perhaps this is the reason for longer hospitalization?). Has it been checked whether the group of all people with reinfection in the database without vaccination differs from those with breakthrough infection?

The paper would have benefited if the results had been supported by, for example, serological data or information on whether reinfection was with the same or a different virus variant.

The authors wrote in line 108-110:" The condition for cases and controls was the presence of antibodies for SARS-CoV-2 by the means of vaccination or prior infection. Patients aged 18 years with proven SARS- CoV-2 infection as determined by positive RT-PCR test findings recorded in our database within the time period specified were eligible for inclusion".

Is there any data available on the SARS-CoV-2 serology of the patients, and if so, what were the antibody levels in the vaccinated breakthrough infection group vs. the reinfected group in relation to the time of infection/reinfection? Did all patients have antibodies after vaccination or primary infection?

In line 154 - I understand that time is an average, please add SD.

Author Response

Dear reviewer,

We all appreciate your feedback and the time taken to evaluate our manuscript. In order to improve our paper, we made the following edits based on your advice, in addition to the other reviewers:

  1. Indeed, we missed to report this important data. In the population of 2041 patients infected with SARS-CoV-2 during the study period, there were only 62 breakthrough cases. However, we do not know the percentage of breakthrough cases (being diagnosed with COVID-19 after receiving the full vaccination scheme – at the time of study there were 2 doses required), since all the other 1979 patients in our records were unvaccinated. Also, it is important to mention that these 62 patients were the only ones vaccinated who required hospital admission, therefore the number of breakthrough cases might be much higher in the general population, but they were either having no symptoms at all, or a very mild form of infection that was not even taken seriously. Also, the vaccination campaign in Romania commenced in January 2021 for the general population. We attempted to describe this concern in the study limitations section (lines 242-246.
  2. In the total of 2041 patients with SARS-CoV-2, 1979 were not vaccinated, and 168 (8.5%) were cases of reinfection. From this number of 168 cases of reinfection, we matched 62 of them by age and gender with the breakthrough cases. This information was added at lines 106-108.
  3. Although the quoted studies describe vaccines as providing better neutralization than natural infection does, our data shows that natural immunity offers a short-term similar protection with a second dose of mRNA vaccine, as mentioned in our conclusions (lines 28-30 and Conclusions section). We believe this “short-term” similar protection is the key here, as vaccines might have a stronger protection for the long-term, which we did not study here.
  4. This is a very good question. However, we do not own this information regarding the severity of initial infection in the cases of reinfection admitted to our hospital. First of all because there isn’t a shared database between all COVID departments in our country, or we do not have the rights to access other departments. Secondly because only moderate and severe cases are considered for hospital admission. Also, the majority of these patients in the reinfection group were admitted to another COVID department. Lastly, the proof that these patients with reinfection were indeed real cases of reinfection was a paper confirmation of a RT-PCR or antibody test, regardless of date of the tests. Please let us know if you consider any of these explanations relevant to be included in the discussion section.
  5. Regarding the selection criteria for cases and controls, we did not compare with existing literature, but we considered age and gender matching to remove the confounding effect of these two factors. We don’t know the algorithm behind case-matching, but we used the “Case-Control Mathcing” function in the SPSS software that was used for data analysis (we added this to the Materials and methods – lines 141-142). Indeed, the reason for longer hospitalization in the reinfection cases was longer than breakthrough cases, and this result can be biased by having significantly more comorbid conditions. We decided to add this to the discussions section (lines 208-209).
  6. Indeed, the paper would have benefited if the results had been supported by serological data or information on whether reinfection was with the same or a different virus variant. We decided to specify this in the study limitations section (lines 249-250).
  7. As we specified above at number 4, the patients brought a confirmation of prior infection by RT-PCR or antibody test. However we can not identify at this point what was the level of antibodies at that time, moreover since not all of them had an antibody test.
  8. Line 156: we added the standard deviation to report the average time.

In addition, we made several other edits based on reviewers’ comments:

Line 47:  we added the beta variant as the reference for vaccination effectiveness.

Lines 250-253: we added that our study rolled during the Beta and Delta waves.

Lines 28 and 253: we added to the abstract and conclusions the fact that our findings are best correlated in case of infection with the Beta or Delta SARS-CoV-2 variants.

Best regards

Reviewer 2 Report

I thank the authors for the interesting study which I read with interest, 

The manuscript is well written, and of much interest. however, some points need to be discussed or changed:

  1. introduction: treatment for COVID19 is now available. Furthermore the efficacy of 95% was during the beta variant, certainly not with omicron 
  2. methods: the study was conducted during the beta and delta variants, however, that's to say that its relevance for the omicron variant may be unclear
  3. discussion: as mentioned before, the discussion and conclusion relevance to the omicron variant is unclear, which is a major drawback for the study

Author Response

Dear reviewer,

We all appreciate your valuable feedback and the time taken to evaluate our manuscript. In order to improve our paper, we made the following edits based on your advice:

Line 47:  we added the beta variant as the reference for vaccination effectiveness.

Lines 250-253: we added that our study rolled during the Beta and Delta waves.

Lines 28 and 253: we added to the abstract and conclusions the fact that our findings are best correlated in case of infection with the Beta or Delta SARS-CoV-2 variants.

In addition, we made several other edits based on reviewers’ comments:

  • In the total of 2041 patients with SARS-CoV-2, 1979 were not vaccinated, and 168 (8.5%) were cases of reinfection. From this number of 168 cases of reinfection, we matched 62 of them by age and gender with the breakthrough cases. This information was added at lines 106-108.
  • Lines 141-142: We added how case-matching was performed by using “Case-Control Mathcing” function in the SPSS software that was used for data analysis.
  • Lines 208-209: We added that the reason for longer hospitalization in the reinfection cases was longer than breakthrough cases can be biased by having more comorbid conditions.
  • Lines 249-250: We added that the paper would have benefited if the results had been supported by serological data or information on whether reinfection was with the same or a different virus variant.

Best regards

Round 2

Reviewer 1 Report

Thank you for your answers and explanations to my questions, which I accept.